# DISENTANGLING ACTION SEQUENCES: FINDING COR-RELATED IMAGES

## ABSTRACT

Disentanglement is a highly desirable property of representation due to its similarity with human's understanding and reasoning. This improves interpretability, enables the performance of down-stream tasks, and enables controllable generative models. However, this domain is challenged by the abstract notion and incomplete theories to support unsupervised disentanglement learning. We demonstrate the data itself, such as the orientation of images, plays a crucial role in disentanglement instead of the ground-truth factors, and the disentangled representations align the latent variables with the action sequences. We further introduce the concept of disentangling action sequences which facilitates the description of the behaviours of the existing disentangling approaches. An analogy for this process is to discover the commonality between the things and categorizing them.

Furthermore, we analyze the inductive biases on the data and find that the latent information thresholds are correlated with the significance of the actions. For the supervised and unsupervised settings, we respectively introduce two methods to measure the thresholds. We further propose a novel framework, fractional variational autoencoder (FVAE), to disentangle the action sequences with different significance step-by-step. Experimental results on dSprites and 3D Chairs show that FVAE improves the stability of disentanglement.

## 1 INTRODUCTION

The basis of artificial intelligence is to understand and reason about the world based on a limited set of observations. Unsupervised disentanglement learning is highly desirable due to its similarity with the way we as human think. For instance, we can infer the movement of a running ball based on a single glance. This is because the human brain is capable of disentangling the positions from a set of images. It has been suggested that a disentangled representation is helpful for a large variety of downstream tasks (Schölkopf et al., 2012; Peters et al., 2017). According to Kim & Mnih (2018), a disentangled representation promotes interpretable semantic information. That brings substantial advancement, including but not limited to reducing the performance gap between humans and AI approaches (Lake et al., 2017; Higgins et al., 2018). Other instances of disentangled representation include semantic image understanding and generation (Lample et al., 2017; Zhu et al., 2018; Elgammal et al., 2017), zero-shot learning (Zhu et al., 2019), and reinforcement learning (Higgins et al., 2017b). Despite the advantageous of the disentangling representation approaches, there are still two issues to be addressed including the abstract notion and the weak explanations.

**Notion** The conception of disentangling factors of variation is first proposed in 2013. It is claimed in Bengio et al. (2013) that for observations the considered factors should be explanatory and independent of each other. The explanatory factors are however hard to formalize and measure. An alternative way is to disentangle the ground-truth factors (Ridgeway, 2016; Do & Tran, 2020). However, if we consider the uniqueness of the ground-truth factors, a question which arises here is how to discover it from multiple equivalent representations? As a proverb "one cannot make bricks without straw", Locatello et al. (2019) prove the impossibility of disentangling factors without the help of inductive biases in the unsupervised setting.

**Explanation** There are mainly two types of explanations for unsupervised disentanglement: information bottleneck, and independence assumption. The ground-truth factors affect the data

independently, therefore, the disentangled representations must follow the same structure. The approaches, holding the independence assumption, encourage independence between the latent variables (Schmidhuber, 1992; Chen et al., 2018; Kim & Mnih, 2018; Kumar et al., 2018; Lopez et al., 2018). However, the real-world problems have no strict constraint on the independence assumption, and the factors may be correlative. The other explanation incorporates information theory into disentanglement. Burgess et al.; Higgins et al.; Insu Jeon et al.; Saxe et al. suggest that a limit on the capacity of the latent information channel promotes disentanglement by enforcing the model to acquire the most significant latent representation. They further hypothesize that the information bottleneck enforces the model to find the significant improvement.

In this paper, we first demonstrate that instead of the ground-truth factors the disentangling approaches learn actions of translating based on the orientation of the images. We then propose the concept of disentangling actions which discover the commonalities between the images and categorizes them into sequences. We treat disentangling action sequences as a necessary step toward disentangling factors, which can capture the internal relationships between the data, and make it possible to analyze the inductive biases from the data perspective. Furthermore, the results on a toy example show that the significance of actions is positively correlated with the threshold of latent information. Then, we promote that conclusion to complex problems. Our contributions are summarized in the following:

- We show that the significance of action is related to the capacity of learned latent information, resulting in the different thresholds of factors.

- We propose a novel framework, fractional variational autoencoder (FVAE) to extracts explanatory action sequences step-by-step, and at each step, it learns specific actions by blocking others' information.

We organize the rest of this paper as follows. Sec.2 describes the development of unsupervised disentanglement learning and the proposed methods based on VAEs. In Sec.3, through an example, we show that the disentangled representations are relative to the data itself and further introduce a novel concept, i.e., disentangling action sequences. Then, we investigate the inductive biases on the data and find that the significant action has a high threshold of latent information. In Sec.4, we propose a step-by-step disentangling framework, namely fractional VAE (FVAE), to disentangle action sequences. For the labelled and unlabelled tasks, we respectively introduce two methods to measure their thresholds. We then evaluate FVAE on a labelled dataset (dSprites, Matthey et al. (2017)) and an unlabelled dataset (3D Chairs, Aubry et al. (2014)). Finally, we conclude the paper and discuss the future work in Sec.5

## 2 UNSUPERVISED DISENTANGLEMENT LEARNING

We first introduce the abstract concepts and the basic definitions, followed by the explanations based on information theory and other related works. This article focuses on the explanation of information theory and the proposed models based on VAEs.

### 2.1 THE CONCEPT

Disentanglement learning is fascinating and challenging because of its intrinsic similarity to human intelligence. As depicted in the seminal paper by Bengio et al., humans can understand and reason from a complex observation to the explanatory factors. A common modeling assumption of disentanglement learning is that the observed data is generated by a set of ground-truth factors. Usually, the data has a high number of dimensions; hence it is hard to understand, whereas the factors have a low number of dimensions, thus simpler and easier to be understood. The task of disentanglement learning is to uncover the ground-truth factors. Such factors are invisible to the training process in an unsupervised setting. The invisibility of factors makes it hard to define and measure disentanglement (Do & Tran, 2020).

Furthermore, it is shown in Locatello et al. (2019) that it is impossible to unsupervised disentangle the underlying factors for the arbitrary generative models without inductive biases. In particular, they suggest that the inductive biases on the models and the data should be exploited. However, they do not provide a formal definition of the inductive bias and such a definition is still unavailable.

## 2.2 INFORMATION BOTTLENECK

Most of the dominant disentangling approaches are the variants of variational autoencoder (VAE). The variational autoencoder (VAE) is a popular generative model, assuming that the latent variables obey a specific prior (normal distribution in practice). The key idea of VAE is maximizing the likelihood objective by the following approximation:

$$\mathcal{L}(\theta,\phi;x,z) = \mathbb{E}_{q_\phi(\mathbf{z}|\mathbf{x})}[\log p_\theta(x|z)] - D_{\mathrm{KL}}(q_\phi(z|x)||p(z)), \tag{1}$$

which is known as the evidence lower bound (ELBO); where the conditional probability $P(x|z), Q(z|x)$ are parameterized with deep neural networks.

Higgins et al. find that the KL term of VAEs encourages disentanglement and introduce a hyperparameter $\beta$ in front of the KL term. They propose the $\beta$-VAE maximizing the following expression:

$$\mathcal{L}(\theta,\phi;x,z) = \mathbb{E}_{q_\phi(\mathbf{z}|\mathbf{x})}[\log p_\theta(x|z)] - \beta D_{\mathrm{KL}}(q_\phi(z|x)||p(z)). \tag{2}$$

$\beta$ controls the pressure for the posterior $Q_\phi(z|x)$ to match the factorized unit Gaussian prior $p(z)$. Higher values of $\beta$ lead to lower implicit capacity of the latent information and ambiguous reconstructions. Burgess et al. propose the Annealed-VAE that progressively increases the information capacity of the latent code while training:

$$\mathcal{L}(\theta,\phi;x,z,C) = \mathbb{E}_{q_\phi(\mathbf{z}|\mathbf{x})}[\log p_\theta(x|z)] - \gamma|D_{\mathrm{KL}}(q_\phi(z|x)||p(z)) - C| \tag{3}$$

where $\gamma$ is a large enough constant to constrain the latent information, $C$ is a value gradually increased from zero to a large number to produce high reconstruction quality. As the total information bottleneck gradually increasing, they hypothesize that the model will allocate the capacity of the most improvement of the reconstruction log-likelihood to the encoding axes of the corresponding factor. However, they did not exploit why each factor makes different contributions to the reconstruction log-likelihood.

## 2.3 OTHER RELATED WORK

The other dominant direction initiates from the prior of factors. They assume that the ground-truth factors are independent of each other, and a series of methods enforce the latent variables have the same structure as the factors. FactorVAE (Kim & Mnih, 2018) applies a discriminator to approximately calculate the total correlation (TC, Watanabe (1960)); $\beta$-TCVAE (Chen et al., 2018) promotes the TC penalty by decomposing the KL term; DIP-VAE (Kumar et al., 2018) identifies the covariance matrix of $q(z)$.

## 3 DISENTANGLING ACTION SEQUENCES

For machines and humans, disentangling the underlying factors is a challenging task. For instance, there are more than 1,400 breeds of dogs in the world, and it seems impossible for an ordinary person to distinguish all of them just by looking at their pictures. The challenge in disentangling the underlying factors is mainly due to the complexity of establishing relationships without supervision, where the corresponding models should contain some level of prior knowledge or inductive biases. However, it is possible to determine the differences without having extensive knowledge. For example, one may mistakenly identify the breed of a dog by looking at a picture but is almost impossible to misrecognize a dog as a cat. Therefore, in practice, discovering the differences or similarities is often a much easier task than that of uncovering the underlying factors, and it does not also need much prior knowledge. One may conclude that discovering the commonalities between things is an important step toward disentanglement.

### 3.1 ACTION SEQUENCES

**Action** *The continuous set of images over a certain direction.*

The observed data consists of a set of actions such as rotation, translation, and scaling. Such actions are **meaningful** and **underlying**. Therefore, the goal of disentanglement learning is to separate the underlying actions of observation. Usually, the continuous actions are infeasible for the machine learning system, and we have to convert them into some discrete and consistent sequences.

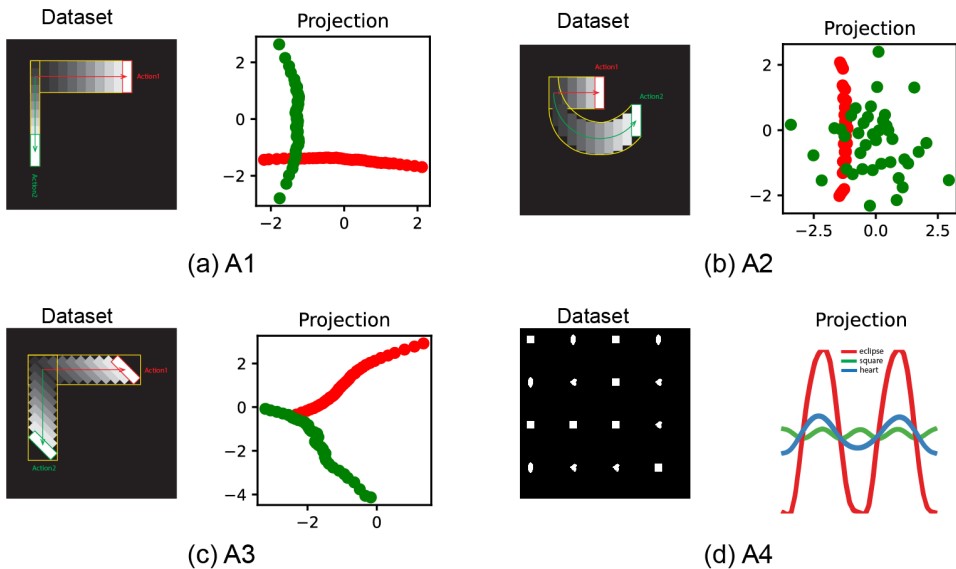

Figure 1: Examples of the influences of the data on the learned representations. (a) Translation perpendicular to the coordinate axis. (b) Translation along the polar axis. (c) Translation of rotated images along the coordinate axis. (d) The learned presentation of three shapes on dSprites. Each line denotes the projections of the rotation angles into the latent variables w.r.t. three shapes.

Given a dataset $\mathbb{S}$, we assume it consists of parameterized action sequences, and each action sequence is a subset of the dataset. We can model a parameterized action sequence as:

$$\mathbb{S}^i(m_{j,j \neq i}) = T_{m_{j,j \neq i}}(\{\mathbf{t}\}),\tag{4}$$

where $\mathbb{S}^i \in \mathbb{S}$ denotes an action sequence , $i$ denotes the type of actions, $m$ denotes a parameter vector, $T$ denotes a transformation, $\mathbf{t}$ denotes the step of the action. The procedure is similar to the latent traversal in Higgins et al. (2017a). Differently, action sequences describe both the real observed data and the reconstructed data: For the real observed data, $m$ is the ground-truth factor vector, and $T$ is the generating function; For the reconstructed data, $m$ is the latent variable vector, and $T$ is a neural network. For clarity, we denote generating actions as sequences of images controlled by the ground-truth factors and action sequences as the approximation of these actions generated by a neural network.

However, the neural networks are flexible enough to approximate any actions, and it's tricky to infer and learn the generating actions. The marvelous thing about disentanglement learning is that the models seem to learn these actions unsupervisedly. A popular view of disentanglement believes the ground-truth factors should be separated into independent latent variables Locatello et al. (2019); Chen et al. (2018). That means the learned action sequences have to match the generating actions precisely. However, (Rolinek et al., 2019) shows that the VAEs conduct a PCA-like behavior, and the models prefer to extract the principal component. Hence, the models may select some action sequences having more considerable variations and learn the significant action sequences rather than the generating actions. Using VAEs, minimizing the objective increases the overlap between the posterior distributions across the dataset Burgess et al. (2018), and it then leads to the learned action sequences showing some internal relationships. Although (Locatello et al., 2019) suggest that the inductive biases on the models and the data should be exploited, the current researches focus on the role of the models on disentanglement Burgess et al. (2018); Chen et al. (2018); Rolinek et al. (2019).

## 3.2 INDUCTIVE BIAS ON THE DATA

We believe the data itself containing the vital information for disentanglement. In other words, *keeping the applied model the same, the learned representations change corresponding to some modifications on the data.* Therefore, there exist some data clues inside the data guiding the models to disentangle. We create a toy dataset family— each dataset contains 40x40 images which are generated from an original image of an 11x5 rectangle by translating on a 64x64 canvas. Each image on the dataset has a unique

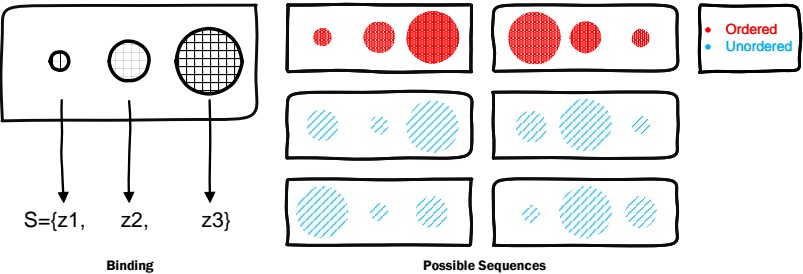

Figure 2: An example of action sequences. There are three images with differing sizes on the dataset (left box), and the AE learns a representation with one dimension of latent space (the double-arrow lines). So the total number of possible sequences is six. Two of them are meaningful, and the others are somewhat random.

label (position of X, position of Y) to describe the ground-truth factors. In this dataset family, there are two variables: the orientation of the rectangle and the way to determine the two factors. There are infinite solutions to determine these two factors; the polar coordinate system and the Cartesian coordinate system are the most common solution. We then create a baseline dataset, A1, with a horizontal rectangle in the Cartesian coordinate system and obtain its variants. A2 differs in the positions determined by the polar coordinate system, and A3 differs in the orientation (45 degrees) of the images. A4 uses the results of dSprites, and we only show the rotation parts w.r.t. three shapes. For the **experiment settings** in this section, we choose the well-examined baseline model, $\beta$-VAE ($\beta=50$), and the other settings such as the backbone network, learning rate, and optimizer refers the settings in Locatello et al. (2019).

As it is shown in Fig. 1, we visualize the learned representations in the latent space. One can see that A1 and A3 have the same generating actions but different in the orientation of the rectangles, and the difference of the rectangles causes the rotation of the learned representations; the model fails to learn the generating actions, however, the learned action sequences are explanatory and similar to A1's (see in A.4). We argue that current approaches don't guarantee to separate the ground-truth factors into isolating dimension (the same conclusion as Locatello et al. (2019)), because A2 and A3 fail to learn the generating action sequences. However, the data guarantee the learned action sequences (A1, A3). One can see that the invariant in A1, A2, and A3 is that they have learned two actions moving along the direction of the long side of the rectangle and the orthogonality direction (see in Fig. 10).

### 3.3    SIGNIFICANCE OF ACTION SEQUENCES

It is suggested in Burgess et al. (2018) that the underlying components have different contributions to the objective function. (Rolinek et al., 2019) addressed that the VAEs conduct PCA-like behaviors. However, they didn't exploit the inductive biases on the data explicitly. Therefore, we use entropy to measure the information among this action:

$$H(\mathbb{S})=-\int_{x\in\mathbb{S}}p(\boldsymbol{x})\log(p(\boldsymbol{x}))d\boldsymbol{x}, \tag{5}$$

where $\mathbb{S}$ is an action, $x$ is a image belonging to this action, $p(x)$ is the probability of the image occupation in this action. However, this formula cannot be applied directly. For the discrete situation, we assume each image is a sample from an action distribution which obeys the Gaussian Drstribution $N(\mu,\sigma^2)$, and we use $\bar{X}$ to estimate the action distribution. Hence, we obtain the approximate entropy by sampling:

$$H(\mathbb{S}')=-\frac{1}{N}\sum_{\boldsymbol{x}_i\in\mathbb{S}'}\log(\frac{1}{\sigma\sqrt{2\pi}}\exp^{-\frac{(\boldsymbol{x}_i-\bar{X})^2}{2\sigma^2}}), \tag{6}$$

where $\mathbb{S}'$ is the set of an action, $\bar{X}=\frac{1}{N}\sum_{\boldsymbol{x}_i\in\mathbb{S}'}\boldsymbol{x}_i$.

In this part, we build a new translation family (A5) with two controllable parameters $\theta,L$, where $\theta$ is the orientation of the rectangle, $L$ is the translating distance of the rectangle. A5 has only one factor and indicates an action moving from the left to the right, and $\theta and L$ controls the entropy of this action.

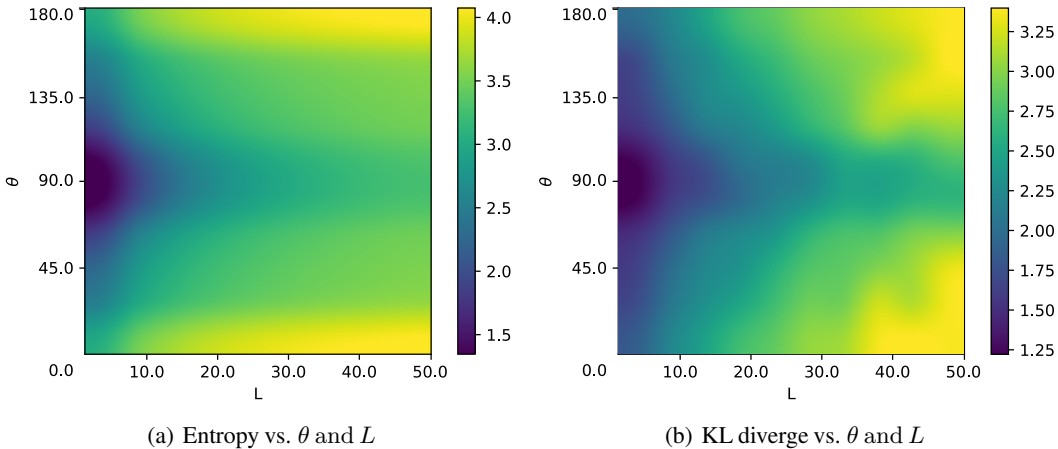

(a) Entropy vs. $\theta$ and $L$        (b) KL diverge vs. $\theta$ and $L$

Figure 3: Comparison between the entropy of actions and KL divergence of latent variables.

As shown in Fig. 3(a), a high value of $L$ means a long traveling area and a small overlap, which has large entropy. If the value of $L$ is small enough, the images in the sequence are almost the same, the action has small entropy or is insignificant. Similarly, the action has more overlap when $\theta = 90$. Note that the maximum for both are reached when $\theta = 0$ or $180$ and $L$ is at its maximum. To measure how much information the model has learned, we select KL divergence to indicate the latent information. The experimental results reveal a similar trend between the entropy and the KL diverge in Fig. 3(b). Therefore, a higher significance of the action ends in more information on the latent variables. In contrast, gradually increasing the pressure on the KL term, the latent information decreases until it reaches zero. One can infer that there exists a critical point that the model can learn information from the action. We call that **the threshold of latent information**. We hypothesize the significance of actions and latent information thresholds are positive correlation (A.5).

## 4 PLAIN IMPLEMENTATION ACCORDING TO THRESHOLDS

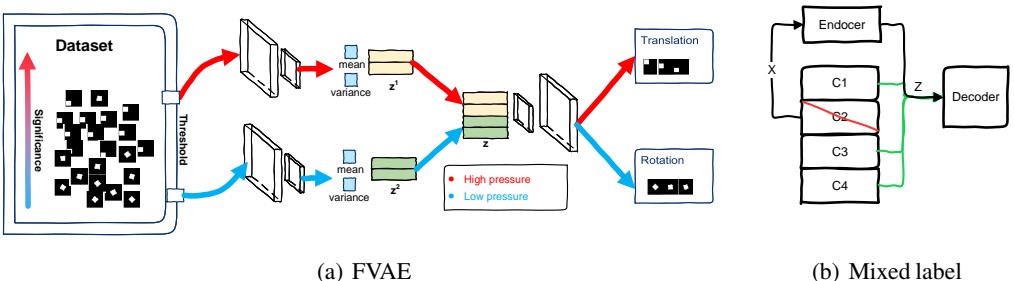

(a) FVAE          (b) Mixed label

Figure 4: (a) The architecture of FVAE. Though the samples distribute in the dataset randomly, they have intrinsic significance. Under a high pressure ($\beta$), the significant actions can pass information alone the red line to itself along, and the information of insignificant actions is blocked. (b) The decoder receives the label information except for the target action.

We have discussed the situation with only one action, and there are various thresholds for different significant actions. Particularly, if $\beta$ is large enough, information of the insignificant actions will be blocked, decaying to a single factor discovery problem. From the modeling perspective, the learning process is similar to a single action learning problem. However, the difficulty of disentanglement is that different kinds of ground-truth actions are mixed, and a single and a fixed parameter $\beta$ is unable to separate them. Therefore, the plain idea is to set different thresholds on the learning phases, and then in

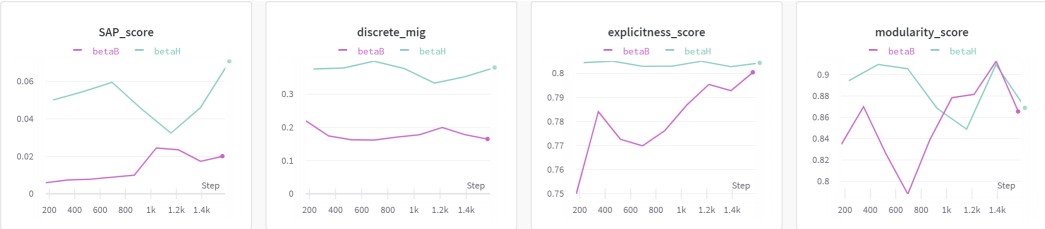

Figure 5: Re-entanglement phonomenon.

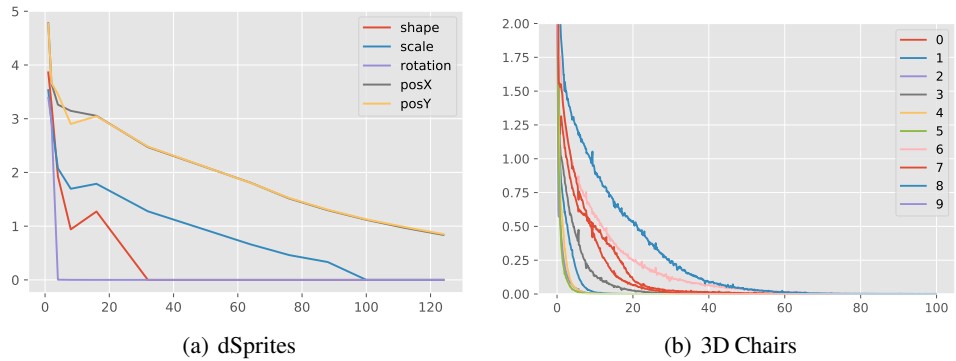

(a) dSprites  (b) 3D Chairs

Figure 6: $\beta$ vs. KL divergence on dSprites (left) and 3D Chairs (right). Each line denotes the dimensional KL diverge over $\beta$ increasing.

each phase, we enforce the model to learn specific actions by blocking information of the secondarily significant actions. We propose a fractional variational autoencoder (FVAE) which disentangles the action sequences step-by-step. The architecture of FVAE is shown in Fig. 4(a). The encoder consists of several groups of sub-encoders, and the inputs of the decoder are the concatenated codes of all sub-encoders. Besides, to prevent re-entangling the learned actions (see in Fif. 5), we set different learning rates for the sub-encoders, which reduces the learning rate for the reset of N-1 groups and prevent the model from allocating the learned codes. The training process of FVAE is similar to the common operation in chemistry for separating mixtures—distillation. To separate a mixture of liquid, we repeat the process of heating the liquid with different boiling points, and in each step, the heating temperature is different to ensure that only one component is being collected.

**Discussion** Although AnnealedVAE follows the same principles as the FVAE, it differs in the interpretation of the effects of beta, and it does not explicitly prevent mixing the factors. Moreover, the performance of AnnealedVAE depends on the choice of hyperparameter in practice Locatello et al. (2019). "A large enough value" is hard to determine, and the disentangled representation is re-entangled for an extremely large C. To address this issue, here we introduce two methods to determine the thresholds in each phase for the labeled and unlabelled tasks.

### 4.1 LABELLED TASK

For the labeled setting, we focus on one type of action and clip the rest of them at first. However, the samples of one action are usually insufficient. For example, there are only three types of shapes on dSprites. Besides, the label information may be corrupted, and only some parts of the dataset are labeled.

To address these issues, we introduce the architecture shown in Fig. 4(b), in which the label information excepted for the target actions are directly provided to the decoder. We evaluate FVAE on the dSprites dataset (involving five actions: translating, rotating, scaling, and shaping). We first measure the threshold of each action, and the result is shown in Fig. 6(a). One can see that the thresholds of translating and scaling are higher than the others. This suggests that these actions are significant and easy to be disentangled. This is in line with the results in Burgess et al. (2018); Higgins et al. (2017a).

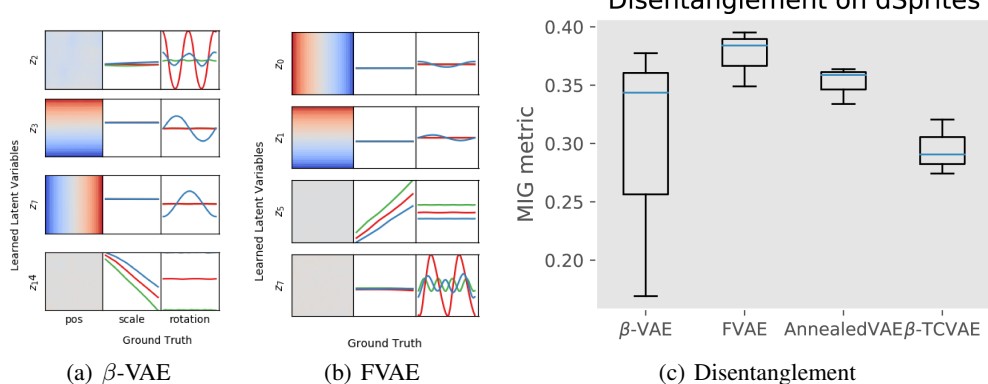

(a) $\beta$-VAE       (b) FVAE       (c) Disentanglement

Figure 7: Comparison between (a) $\beta$-VAE and (b) FVAE. We visualize the average projections of the factors into the most informative latent variable on dSprites (only active units are shown). The colored lines indicate three types of shapes (red: eclipse, green: square, blue: heart). (c) The disentanglement scores of $\beta$-VAE and FVAE (MIG, Chen et al. (2018)).

According to these thresholds, we then arrange three stages for dSprites. At each stage, we set up a bigger number of $\beta$ than the threshold of the secondary action. The pressure on the KL term also prevents the insignificant actions from being disentangled and ensures that the model only learns from the information of the target action. The training of each stage can be found in the Appendix. As shown in Fig. 7(a), the translation factor is disentangled at first easily, while it is hard to distinguish the shape, orientation, and scale. Gradually, scaling and orientation also emerge in order. Nevertheless, it should be noted that the shaping is still hard to be separated. This could be attributed to the lack of commonalities between these three shapes on dSprites and motion compensation for a smooth transition. In other words, in terms of shape, the lack of intermediate states between different shapes is an inevitable hurdle for its disentanglement. Fig. 7 shows a more substantial difference between the $\beta$-VAE and the FVAE. $\beta$-VAE has an unstable performance compared to FVAE, and position information entangles with orientation on some dimension.

## 4.2 UNLABELLED TASK

For the unlabelled setting, we introduce the annealing test to detect the potential components. In the beginning, a very large value for $\beta$ is set to ensure that no action is learned. Then, we gradually decrease $\beta$ to disentangle the significant actions. There exists a critical point in which the latent information starts increasing, and that point approximates the threshold of the corresponding action.

3D Chairs is an unlabelled dataset containing 1394 3D models from the Internet. Fig. 6(b) shows the result of the annealing test on 3D Chairs. One can recognize three points where the latent information suddenly increases: 60, 20, 4. Therefore, we arrange a three-stage training process for 3D Chairs (more details in the Appendix). As shown in Fig. 7(b), one can see the change of azimuth in the first stage. In the second stage, one can see the change of size, and in the third stage, one can see the change of the leg style, backrest, and material used in the making of the chair.

## 5 CONCLUSION

We demonstrated an example of the effects of images' orientation on the disentangled representations. We have further investigated the inductive biases on the data by introducing the concept of disentangling action sequences, and we regarded that as discovering the commonality between the things, which is essential for disentanglement. The experimental results revealed that the actions with higher significance have a larger value of thresholds of latent information. We further proposed the fractional variational autoencoder (FVAE) to disentangle the action sequences with different significance step-by-step. We then evaluated the performance of FVAE on dSprites and 3D Chairs. The results suggested robust disentanglement where re-entangling is prevented.

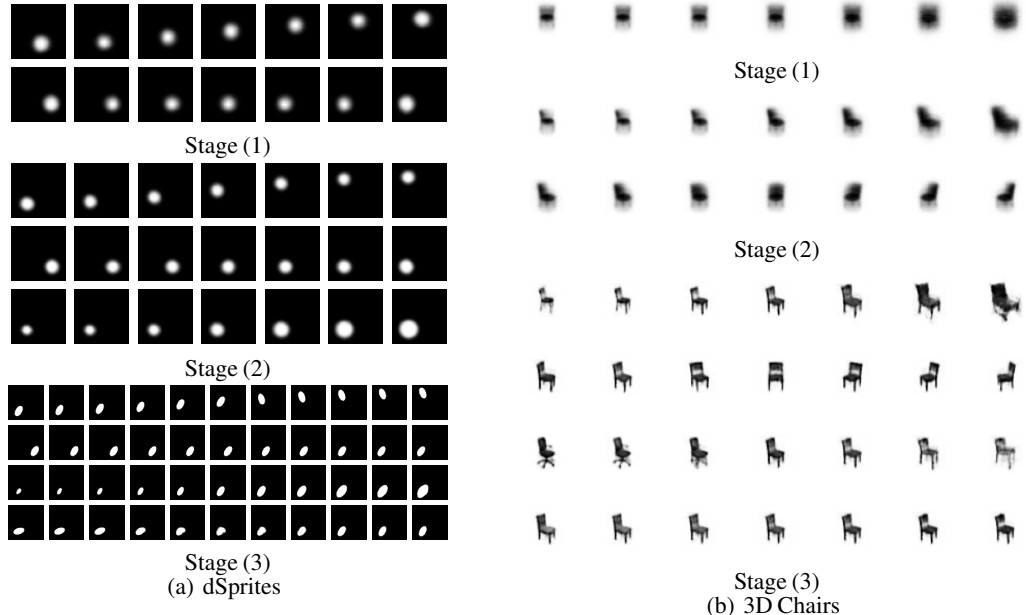

Stage (1)

Stage (2)

Stage (3)

(a) dSprites

Stage (1)

Stage (2)

Stage (3)

(b) 3D Chairs

Figure 8: **FVAE disentangles action sequences step-by-step.** Latent traversals at each stage. The left is the results on dSprites. The right is the results on 3D Chairs.

This paper proposed a novel tool to study the inductive biases by action sequences. However, other properties of inductive biases on the data remain to be exploited. The current work focuses on an alternative explanation for disentanglement from the perspective of information theory. In the future, the influence of independence on disentanglement requires further investigation.

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

| Phase | 1 | 2 | 3 |
|---|---|---|---|
| $E_1$ | **5e-4** | 5e-5 | 5e-5 |
| $E_2$ | 0 | **5e-4** | 5e-5 |
| $E_3$ | 0 | 0 | **5e-4** |
| $\beta_1$ | 100 | 40 | 4 |
| $\beta_2$ | 60 | 20 | 2 |

Table 1: Training settings on dSprites and 3D Chairs. $E_i$ denotes the learning rate of $i$-th group of sub-encoder. $\beta$ denotes the regularisation coefficient before the KL divergence, $\beta_1$ for dSprites, $\beta_2$ for 3D Chairs

| Factor | Threshold |
|---|---|
| Shape | 32 |
| Scale | 100 |
| Orientation | 5 |
| Position X | 120+ |
| Position Y | 120+ |

Table 2: Thresholds of actions on dSprites.

| Factor | Threshold |
|---|---|
| Chair size | 60 |
| Leg style | 20 |
| Swivel | 2 |
| Unknown | 2 |

Table 3: Thresholds of actions on 3D Chairs.

# A APPENDIX

## A.1 DATASETS

## A.2 DSPRITES AND 3D CHAIRS

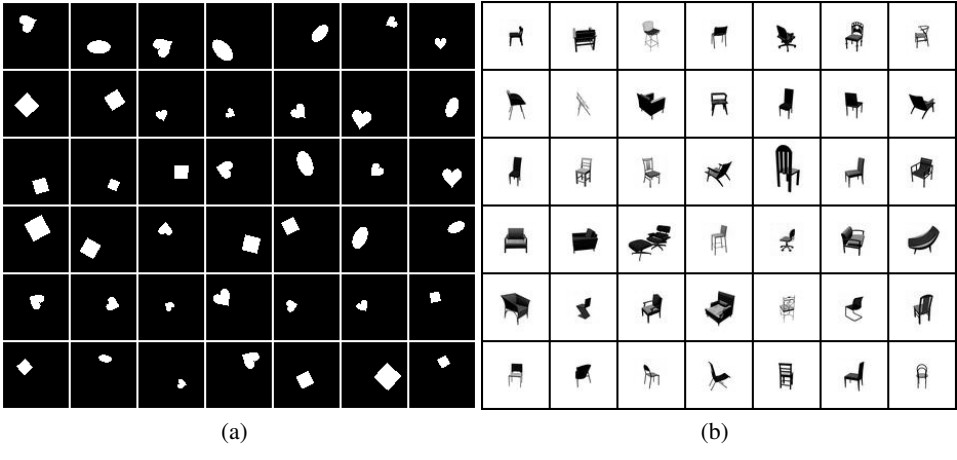

(a)                                             (b)

Figure 9: Real samples from the training dataset.

## A.3 TRAINING DETAILS

The basic architecture for all experiments follows the settings on Locatello et al. (2019). The hyperparameters of our proposed methods are listed in Tab. 1. Tab. 3 and 2 show the measured thresholds of the intrinsic action sequences.

## A.4 LEARNED ACTION SEQUENCES

Fig. 10 is the supplemental results of the experiments on Sec 3.2.

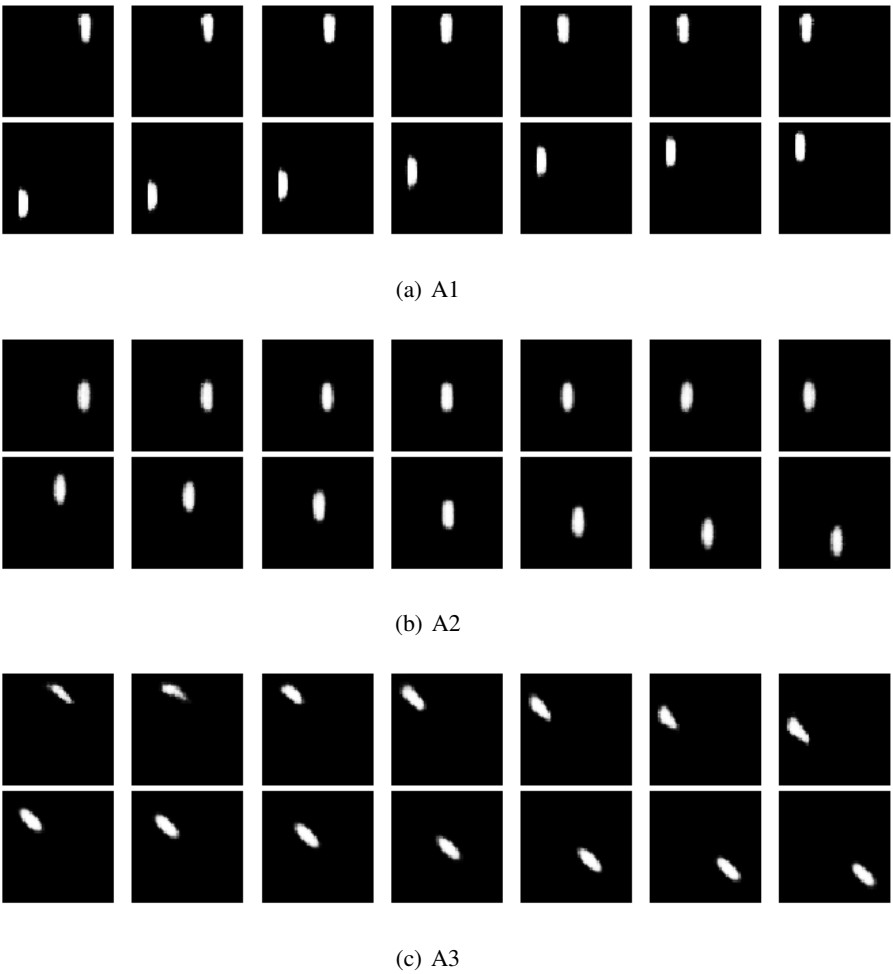

(a) A1

(b) A2

(c) A3

Figure 10: Latent traversal of the trained models on A1, A2, and A3.

## A.5 THE HYPOTHESIS OF KL

We hypothesize the KL diverge is inversely proportional to $\beta$ and positively proportional to $H$:

$$\mathrm{KL} = \frac{H(\mathbb{S})}{\beta^2 + \mathrm{C1}} * \mathrm{C2},\tag{7}$$

where C1,C2 are constant. We examine this in Fig. 11.

## A.6 SAMPLES

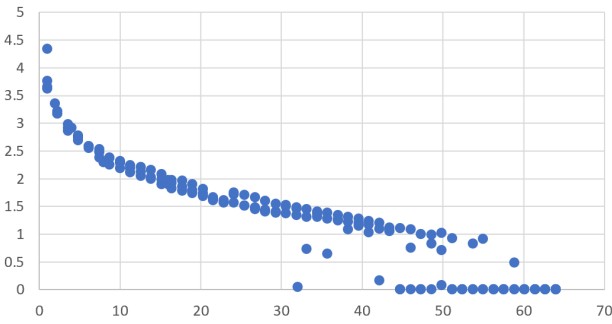

Figure 11: KL vs. $\beta$.

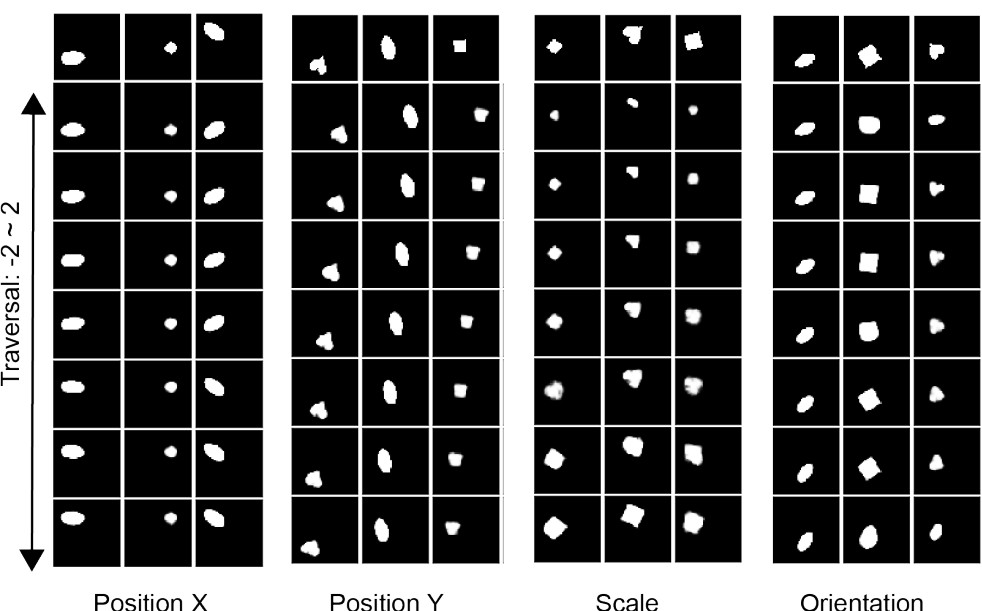

Position X  Position Y  Scale  Orientation

Figure 12: Latent traversal plots for FVAE on dSprites. The top row show the real samples from dSprites.

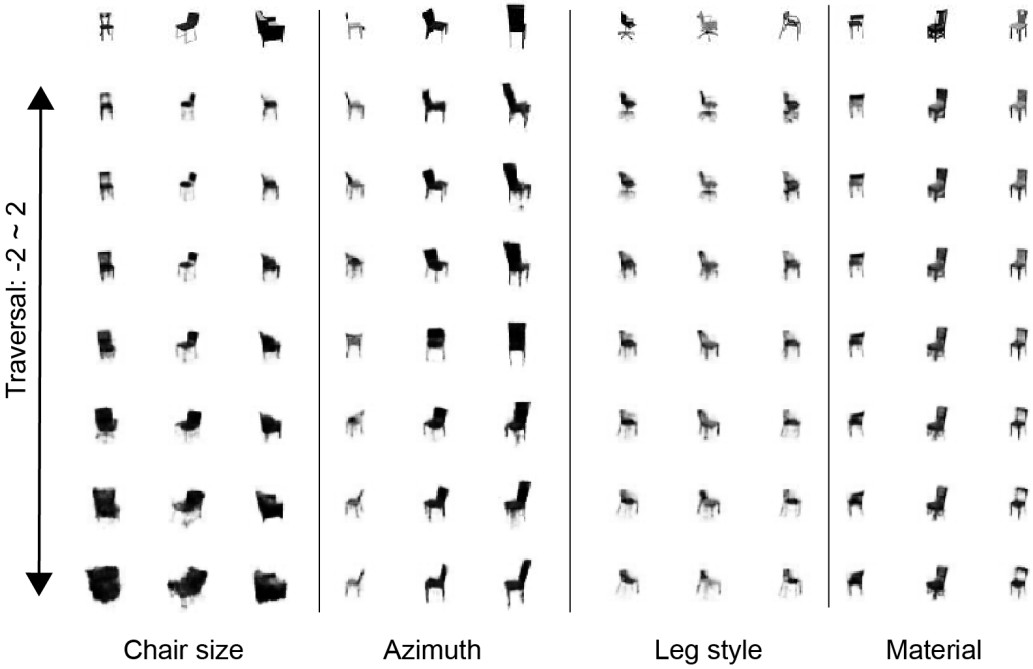

Figure 13: Latent traversal plots for FVAE on 3D Chairs. The top row show the real samples from 3D Chairs.

