# OpenReview forum: "Disentangling Action Sequences: Discovering Correlated Samples"
_ICLR.cc/2021/Conference — Reject_

### Official Review · AnonReviewer1 · 2020-10-23
**Algorithm is not clearly explained and more experiments are needed.**

**Rating:** 2
**Confidence:** 5

**Review:**

Summary:

The authors proposed fractional variational autoencoder (FVAE) for the learning of disentangled representation where the action sequences can be extracted step-by-step. Experiments are shown to illustrate how the algorithm works.

#################


. The authors proposed FVAE but the associated objective function is not introduced explicitly, which is confusing. Is it the same as the objective of \beta-VAE?

. Fig.3: 1) What's the KL divergence here? Is it between the posterior and the prior? 2) It's claimed that the trend of KL divergence is consistent with that of entropy. But it is hard to see from Fig. 3. 3) It is claimed that the significance of action is related to the capacity of learned latent information. Based on Fig. 3, this conclusion is not convincing. Also, Fig. 3 is obtained based on a toy dataset. To claim it as a main contribution, the conclusion needs to be verified on other datasets as well.

. Section 4.1: What's definition of the label here? It's not clear. Is it like the types of shapes on dSprites?

. Section 4.1: The training on dSprites includes two phases: find thresholds and then train different stages. 1) The authors arranged  three stages for dSprites. This seems arbitrary. Why not four or five stages? 2) What's the training objective function of each stage? 3) How are the curves in Fig. 5 derived? More explanation is required.

. Section 4.2: It is claimed that ``One can recognize three points where the latent information suddenly increases: 60, 20, 4.'' This is hard to see from Fig. 5b) as all curves look smooth. Thus, the following three-stage training process is questionable. The training for unlabeled task needs more study.

. The experiments are limited. There are a lot of papers regarding disentangled representation, and the authors only compared with \beta-VAE.

---

> ### Author Response · Authors · 2020-11-25
> **Response to Reviewer1**
>
> Thanks for your detailed feedback and the insightful reviews!
>
> > What's the KL divergence here?
>
> That's right. The KL divergence is the regularization term of the VAE objective.
>
> > KL divergence is consistent with that of entropy.
>
> Basically, the relationship between the KL term and entropy has not yet been verified. We admit that mainly is a hypothesis, though we conduct two experiments in Fig. 3 and 11. However, the current works, such as AnnealedVAE and PCA Directions indicate similar results that different actions have different thresholds. Indeed, such a conclusion can hardly induce; hence, abundant experiments are required. However, the available datasets are insufficient to verify it.
>
> > What's definition of the label here?
>
> The dSprites is an artificial dataset and contains the factors information. The labels are ground-truth factors. Although our method is an unsupervised approach, the calculation of action's entropy needs label information.
>
> > The authors arranged three stages for dSprites.
>
> The direct answer is prior to this dataset. We already know it consists of five factors, and two are similar (posX, posY); the shape is not a good action. Besides, we also provide an annealing test in Sec 4.6 to no prior information case. Increasing the number of stages has little influence on disentanglement, but it does waste the computational resource.
>
> > training objective function
>
> The objective function is beta-VAE. The only difference is the training process.
>
> > How are the curves in Fig. 5 derived?
>
> Each line denotes the dimensional KL diverge over β increasing. For the supervised case, we can name the dimension by its most informative factor. For the unsupervised case, we show their index of dimension.
>
> > Thus, the following three-stage training process is questionable.
>
> For now, FVAE needs the participation of humans. It may not be a bad idea because unsupervised disentanglement learning without inductive biases is impossible. The main purpose of this work is not to propose a competent disentangling method. We focus on the interpretation of why VAEs can disentangle. In this work, we try to provide an insight into combining the data and the representation, and the thresholds could help disentanglement.
>
> Reference:
> - Burgess et al., ”Understanding disentangling in beta-VAE”, NeurIPS 2018.
> - Michal Rolinek; Dominik Zietlow; Georg Martius: Variational Autoencoders Pursue PCA Directions (by Accident), CVPR 2019.

---

### Official Review · AnonReviewer3 · 2020-10-28
**Disentangling action sequences is interesting but more details and experimental results are needed.**

**Rating:** 5
**Confidence:** 4

**Review:**

Summary:
This paper addresses the problem of disentangling representations using Variational Autoencoders. In particular, the authors introduce the concept of disentangling action sequences and propose the fractional variational autoencoder framework to disentangle them step-by-step. To this end, they analyze the inductive biases on the data and define latent information thresholds which are correlated with the significance of the actions.

##################################################################

Strengths:
- The paper tackles the important problem of disentangling representations.
- Overall, the paper is well structured. In particular, the introduction section clearly motivates the problem and summarises existing approaches.
- The idea of disentangling action sequences is interesting as it allows to analyse the inductive bias on the data.

##################################################################

Weaknesses:
- Fractional variational autoencoder (FVAE) proposed in this paper is closely related to the work of Burgess et al. (2018). Although authors include a brief discussion comparing both methods, the main novelty of FVAE (i.e. explicitly avoid mixing the factors and defining thresholds to prevent re-entanglement for extremely high capacity) is still not sufficiently emphasised throughout the paper. Moreover, it would be good to include experimental comparisons to Annealed VAE (for instance in Figure 6) to give more insights on the relevance of the proposed approach.
- Description of the toy dataset family is not easily understood and it would be good to clarify annotations in Figure 1(a). Since Figure 9 of Appendix is clear enough, it might be nice to include it in the main paper to help the reader follow the analysis of the corresponding experiment. In the latter, it is shown that the disentangled representations are not invariant to orientation of rectangles (A1, A3). Here, one can assume that positions x and y and orientation of rectangles contribute differently to reconstruction. Hence, it would be interesting to see the effect of progressively increasing the bottleneck capacity on the obtained representations, as proposed in Burgess et al. (2018). Would this help disentangle position at first then orientation of rectangles?
- While a quantitative analysis has been provided in Figure 6 using the MIG metric to compare FVAE and Beta-VAE, it is still insufficient to make clear conclusions on the performance of the proposed method. Several metrics (e.g. Mutual Information Gap, Modularity, etc.) and evaluation benchmarks have been integrated in DisLib (Locatello et al. (2019)) allowing easy evaluations of disentangling approaches. I recommend using it for further quantitative analysis.
- In Section 2, authors present the concept of disentangling representations and describe the work of Locatello et al. (2019) which shows the necessity of inductive bias to unsupervisedly disentangle the underlying factors. After mentioning that “a formal definition of the inductive bias is still unavailable”, this point would be more clear with some examples, for instance the assumption in Burgess et al. (2018) that Beta-VAE aligns latent dimensions with components that make different contributions to reconstruction.
- In Section 3.1, authors mention that existing models disentangle the ground-truth factors by accident. This would be a little misleading to previous claims on the role of inductive bias on the data (or the model) which allows to achieve disentanglement  Locatello et al. (2019). I suggest more clarification to this point.

---

> ### Author Response · Authors · 2020-11-24
> **Response to reviewer 3.**
>
> Thanks for your detailed feedback and the insightful reviews!
>
> **The difference between Annealed VAE**
> The interpretation of disentanglement: AnnealedVAE argues that the information bottleneck enforces the model to encode "the most significant improvement in data log-likelihood," which leads to disentanglement.  In contrast, we claim that a high regularization penalty on KL divergence prevents the insignificant action sequences from being encoded. In other words, the key to disentanglement is learning information solely.
>
> > Would this help disentangle position at first then orientation of rectangles?
>
> We are sorry for the confusing description that A1-3 have the same two-dimension factors denoting images' position. The orientation of images is a fixed property of the dataset. "The most significant improvement" should have the largest variation, which can also understand from the PCA theory. In fact, the current theories (information bottleneck, PCA Directions) support the results of A1-3. The action sequence moving along the rectangle's short side has the largest variation and improves the log-likelihood the most significantly. Hence, we say disentangling action sequences is a proper description of disentanglement.
>
> > a formal definition of the inductive bias is still unavailable.
>
> Though Burgess and Rolınek indicate the inductive bias, they don't propose a calculation for that.
>
> > existing models disentangle the ground-truth factors by accident.
>
> Rolınek used a similar expression, "Variational Autoencoders Pursue PCA Directions (by Accident)." That means the success of the current approaches (beta-VAE, TC-VAE, AnnealedVAE, DIP-VAE, FactorVAE) mainly contributes to the well-designed dataset. For instance, they fail to disentangle in the cases of A1-3. If the disentangled representation depends on the data, these approaches don't guarantee the disentanglement when facing an unknown dataset.

---

### Official Review · AnonReviewer4 · 2020-10-28
**This paper gives a new kind of comprehension to disentangled representation learning. In this paper, existing unsupervised disentangled methods are trying to obtain disentangled action sequences instead of independent factors. Through the proposed FVAE model, action sequences with different levels of significance can be obtained step by step, and experiment results show that the weight \beta has positive correlation with the significance.**

**Rating:** 6
**Confidence:** 4

**Review:**

Pros:
1. This paper gives a new comprehension of existing unsupervised disentangled representation learning method, regarding it as finding commonality of input data and disentangling action sequence information in factors.
2.  This paper gives a new idea that \beta has positive relationship with action significance and conduct an experiment to validate it.
3.  This paper proposes a new variant of VAE called FVAE which learns the disentangled action sequence step by step.
Cons:
1.  This paper mainly gives descriptions of insights while lacking some formulations to explain the settings and methods better.
2.  Experimental results are not well organized, some axis lack corresponding labels, like Figure 5. Figures are not clear, like numbers in Figure 1 (a).
3.  Some definitions are ambiguous, like x in equation 5.
4.  Some descriptions in the paper are confusing, e.g. “We argue that the factors are not the key to disentanglement since the learned representations are changed while the factors are unchanged (A1, A3), and the learned representations do not change while the factors are changed (A1, A2).” This experiment, from my perspective, shows that the learned factors are disentangled in a particular form which is not consistent with the preset ground truth. And different action sequences are also different factors. Section 3.1 might be described in a more considerate way to show what the experiment results really indicate.
5. There exist some typos in this paper, like “leaned” for “learned”.

Overall review:
This paper gives a new comprehension of existing disentangled representation learning by regarding it as finding disentangled action sequences, which is interesting and has some good insights. However, some ideas should be supported by clearer formulations and some conclusions of experiments are not valid. Moreover, the logic of this paper is a little unclear, and experimental figures are incomplete. With some modifications, this paper could be an excellent paper..

---

> ### Author Response · Authors · 2020-11-25
> **Response to Reviewer4**
>
> Thanks for your detailed feedback and the insightful reviews!
>
> **Ambiguous definitions**
>
> 1. Action : *the continuous set of images over a certain direction.*
> 2. Action sequence: the discrete action or a sequence of sampled images from the action.
> 3. Generating action sequence: the action sequences traversing the ground-truth factors.
> 4. Learned action sequence: the action sequences traversing the latent variables.
> 5. Entropy of action:
>
> $$H(S') = - \frac{1}{N}\sum_{x_i \in S'} \mathrm{log}
>         (\frac{1}{\sigma \sqrt{2 \pi}} \exp^{-\frac{(x_i-\bar{X})^2}{2\sigma^2} })$$
> where $S'$ is the set of an action, $x$ is the sampled images form this action, $\bar{X}$ is the mean of the action.

---

### Official Review · AnonReviewer2 · 2020-10-28
**Interesting model idea w.r.t. annealing capacity of latent representation, but relation to competing SOTA approaches requires more clarity**

**Rating:** 4
**Confidence:** 4

**Review:**

### Summary:
In this submission, a common modelling assumption for unsupervised disentanglement is challenged: that the disentangled representation follows the independence structure of the underlying (data generating) factors. Instead, the paper proposes to consider *action sequences* which describe how datapoints are interrelated. The paper provides evidence that the capacity of the latent representation (controlled by Lagrange parameter beta in beta-VAE related models) is related to the significance of particular action sequence for disentanglement. To leverage this insight, the fractional VAE (FVAE) is proposed, consisting of several sub-encoders and different training stages. The disentangling properties of the FVAE is demonstrated on the dSprites and 3D chairs datasets, with the FVAE performing favourably to the beta-VAE w.r.t. the Mutual Information Gap (MIG) disentanglement metric on dSprites.

### Strengths:
- Novelty / relevance: The submission addresses the important topics of inductive biases and disentangling factors in learning disentangled representations and suggest the interesting and novel concept of action sequences which seems to be related to the general idea of uncovering symmetries with deep latent variable models. In particular, the annealing approach with respect to the KL-divergence Lagrange parameter beta in the FVAE setting to separate “significant modes” might pose a relevant insight useful in other related approaches and to a more broader audience.

### Weaknesses:
- Technical quality / significance: The submission mentions the similarities to approaches like AnnealedVAE by Burgess et al. and qualitatively discusses differences and relates some results to this competing approach, but an empirical evaluation of the proposed approach to the competing method is missing. This is quite important, as the technical details of annealing the capacity of the latent representation seem very much alike. Also comparing the disentangling scores to more state-of-the-art approaches like FactorVAE (Kim and Minh) would be important. The evaluation is solely done with respect to the beta-VAE which might not be the most relevant competitor here. For instance, figure 3 in Burgess et al. reports a similar finding as provided in figure 7a in the submission, i.e. controlling the information capacity disentangles first positional / translational factors, then scale and then orientation / shape. Therefore, it is difficult to assess the validity of the claims of the proposed approach and whether a significantly different contribution than in Burgess et al. is made.
- Figure 5a suggests for dSprites that position/translation, scale and shape are the relevant actions in that order. However, the result in figure 7a suggest, that first translation, then scale and lastly rotation are gradually disentangled which seems to contradict the first result in figure 5a. Shouldn’t these be the same?
- Figure 6a and 6b are not explained or discussed and their interpretation is not clear. A reader might be familiar with similar plots e.g. in the paper by Higgins et al., but still the key insight should be stated somewhat more clearly in the paper.
- Clarity: At times it is difficult to follow the presentation of the content in the paper and in some cases I find it hard to follow the statements and conclusions. For instance:
- Toy example in section 3.1, especially last paragraph: I believe the interpretation of the results in figure 1 requires a little bit more explanation. As I understand it, the ground-truth factors here are the positions X, Y of the rectangles. The dataset provides the variables (i) orientation of the rectangle, (ii) coordinates in either Cartesian or polar coordinate system. I do not quite follow the statement that *“[…] learned representations are changed while the factors are unchanged (A1, A3), and the learned representations do not change while the factors are changed (A1, A2).”* In case (A1, A3) I would say the latent representation is the same up to permutation of coordinate axes / rotations, which is inherent to VAE / PCA approaches. I.e. the meaning of the axes would be still the same (up to these transformations). Therefore, I am also not quite sure about the statement: *“As we have shown in Sec. 3.1, the orientation of the rectangle can affect the direction of the disentangled representation”* (p. 5). The “direction” of the representation is less relevant, as the interpretation of the axes is still the same. However, I might miss the point which is tried to be made here. Could the authors comment on that?
- Definition of action sequence, section 3.2: The paper tries to motivate “action sequences” but in my opinion the notion remains somewhat unclear in an abstract setting. A “meaningful action sequence” is defined as *“a sequence / ordered permutation of elements from a subset of the dataset, which reveals the relationship among the elements”*, with elements being images here. In a simple example as scaling or translating objects, this notion and the distinction to “ground-truth factors” might be clearer. However, in more complex / less structured examples, say images of faces, the difference between “action sequence” and “ground-truth factors” is not very clear to me. The paper suggests for a more formal definition to consider Higgins et al. but, in order to be self-contained and clear, a more explicit and formal definition of this notion is required in the paper, in my opinion. Could the authors maybe provide a more formal definition?

### Additional Feedback:
- Page 6 and figure 3: *“Please note that the maximum for both are reached when theta=90 and L is at its maximum.”* Figure 3 suggests that the maximum (yellow region) is reached for large L and theta close to 0 or about 180. It seems that there is a discrepancy between the description and the figure.
- Figure 5, page 7: In 7b the legend specifies integers, but it is not clear, what these integers encode. And is it maybe *“KL divergence vs beta”* (-> *”y against x”*) in the caption?

- Abstract (p. 1): I would suggest rephrasing the following sentence:  *“We demonstrate the data itself, such as the orientation of images, plays a crucial role in disentanglement and instead of the factors, and the disentangled representations align the latent variables with the action sequences.”*
Maybe get rid of the first *“and”* as well as making clear what *“factors”* (maybe rather *“ground-truth / separating factors”*?) are meant. On the first read, this sentence was quite confusing to me.
- Introduction (p. 1): Second sentence, *“thinking”* -> *“think”*.
- Introduction (p. 1): Third sentence, *“[…] single glance this is because […]”* -> *“[…] single glance. This is […]”*.
- Introduction (p. 1): Notion paragraph first word, *“the”* -> *“The”*.
- Introduction (p. 1): Notion paragraph, *“[…] a question arise here is […]”* -> *“[…] a question which arises here is: […]”*.
- Figure 1, caption: *“leaned”* -> *“learned”*.
- Section 3.3, incomplete sentence after equation 5 or unnecessary *“,”*.
- Section 4, page 6: *“[…] leading to the disentangling process decays […]”* -> *”[…] decaying […]”*
- Section 4, page 6: *“[…] targeted action into the leaned codes.“* -> *“[…] learned […]”*
- Figure 7, page 8: Full stop *“.”* missing in the last sentence of the caption.

### Recommendation:
In general, the paper deals with relevant issues in learning disentangled representations and provides interesting tools to address some of these aspects. In particular, the annealing procedure in the FVAE is potentially a relevant contribution. However, the relation to similar approaches is not evaluate adequately, in my opinion, which makes it difficult to assess the justification of some claims. Also, a careful revision of the submission seems advisable which might clarify some of the aspects raised above. In the current form, I believe that the paper is not ready for publication and I would rather see this submission rejected. Nevertheless, I am willing to reconsider my rating if the authors are able to address some of the concerns and questions raised above.

### Post-Rebuttal:
I want to thank the authors for their responses and clarifications. I think the revision already improved the quality of the submission quite a bit. However, I still believe that there are some aspects which need a better presentation and clearer discussion.

For example, a more direct discussion and (empirical) comparison to other approaches like AnnealedVAE is necessary, as also other reviewers pointed out, to justify the points made (qualitatively) in the paper. The added results in figure 6c already provide results in that direction.

I appreciate the clarifications in the notions of action and action sequence. Although I agree that the notions are comprehensible in the toy example and dSprites setting, I still think that the point I raised in my initial review applies. In order to provide a well-defined notion a more formal definition is required. To me it is still unclear what an action sequence in the case of e.g. images of faces should be.

I genuinely believe that the proposed approach might pose a relevant contribution but the paper lacks an adequate presentation at the moment, in my opinion. Therefore, I stand with my initial recommendation that this submission is not ready for publication and I endorse rejecting the paper. However, I would like to encourage the authors to do a major revision taking the issues raised by the reviewers into consideration and to submit again.


### References:
- Higgins et al., “beta-VAE: Learning basic visual concepts with a constrained
variational framework”, ICLR 2017.
- Kim and Mnih, “Disentangling by factorising”, ICML 2018.
- Burgess et al., ”Understanding disentangling in beta-VAE”, NeurIPS 2018.

---

> ### Author Response · Authors · 2020-11-24
> **Response to Reviewer 2**
>
> Thanks for your detailed feedback and the insightful reviews! We feel sorry about failing to show the comprehensive results of our work. We hope our responses correctly answer your concerns.
>
> **The different contributions**
> 1. The interpretation of disentanglement: AnnealedVAE argues that the information bottleneck enforces the model to encode "the most significant improvement in data log-likelihood," which leads to disentanglement.  In contrast, we claim that a high regularization penalty on KL divergence prevents the insignificant action sequences from being encoded. In other words, the key to disentanglement is learning information solely.
> 2. As far as we know, we are the first to define the inductive biases on the data and associate it with disentanglement. Though Burgess and Rolınek indicate something similar, we give a former and explicit definition.
> 3. We improve disentanglement by solving re-entanglement.
>
> **Explanation of toy examples** We want to show some evidence of inductive biases on the data in this part. The learned action sequences should match the generating action sequence precisely for the popular view of disentanglement learning. However, the experimental results of A2 and A3 reveal that the current disentangling approach learns a significant action sequence or the principal component on the data.  We will update this figure for easy understanding. A1 and A3 have the same generating action, but the model learns two different action sequences. In contrast, A1 and A2 have different generating actions, but the model learns similar action sequences.
>
> **Contradiction** The orders should be the same if they are all actions.  However, there are only three types of shape, eclipse, square, and heart.  We don't feel surprised by this result because *shape* is not an action like others having internal frames. An action should consist of a series of continuous images.
>
> **Notion** Action: The continuous set of images over a certain direction.
>
> This notion denotes the real action in reality, i.e., a ball falls. However, the action is infeasible for the machine, and we have to sample the discrete action as *an action sequence*. Therefore, a subset of the dataset varying one factor is an action sequence. We call the action sequences generated by the ground-truth factors *generating actions* or just *actions*, and the reconstructed sequences by the decoder *learned action sequences* or just *action sequences*.
>
>
> Reference:
> - Higgins et al., “beta-VAE: Learning basic visual concepts with a constrained variational framework”, ICLR 2017.
> - Burgess et al., ”Understanding disentangling in beta-VAE”, NeurIPS 2018.
> - Michal Rolinek; Dominik Zietlow; Georg Martius: Variational Autoencoders Pursue PCA Directions (by Accident), CVPR 2019.
> - Francesco et al., Challenging common assumptions in the unsupervised learning of disentangled representations. In 36th International Conference on Machine Learning, ICML 2019.

---

### Official Review · AnonReviewer5 · 2020-11-09
**Review 5**

**Rating:** 3
**Confidence:** 4

**Review:**

Manuscript Summary
====

This paper constructs a disentanglement problem from a temporally causal view, where data are observed in sequences, and where actions cause those observations to change as the sequence progresses in time. Their stated objective is to recover the specific actions and their parameters (e.g. rotation and translation, and their magnitudes/signs).

The authors thus construct a "Fractional VAE" (FVAE), and then construct sequences from Dsprites and Chairs based on their statement of the problem.

Initial Decision (from this reviewer), Review, and Reasoning
====

I think this paper should be rejected; were this a journal, I would suggest at least major revision.

Overall the concept of bringing temporal causality (which for some cases *is* valid as the causal diagram/frame) into the disentanglement problem statement is a good idea. However, after that point in the manuscript, I cannot understand what has done. For example, section 2 is a restatement of previous work, and section 3 begins with an explanation of the dataset construction. Section 3.3, section 4, and Figure 4A I think describe the paired asymmetric autoencoder method, but a sparse few paragraphs are given at this point. What they _do_ describe is a set of "sub-encoders" with varying compression rates $\beta$. However, beyond varying the rates, it's not clear how particular "ground truth" factors (e.g. $\theta, L$) can be selected for and locked in to specific latent factors in an unsupervised manner, or even if this should happen.

By varying $\beta$ we receive different amounts of information in the representation, but how can we ensure across "learning phases" disentanglement? Further, if these KL divergences are set to different $\beta$, this means we don't have a divergence for the joint representation? (the concatenation of the sub-encoders) So how can we ensure that these are disentangled themselves? While successively learned encodings would optimally not include previously encoded data, why would these encoders learn separate concepts instead of coarse grained representation with all concepts to successively finer representations (or refinements to those coarse grained representations) with each successive sub-encoder.  Or are these separate phases repeated?

Perhaps these questions have answers in the positive, but they should be answered by the manuscript.

I further cannot make a connection between the actions sequences and the training methods/arch. I think I have understood both (...save for the above highlighted problems), but I cannot understand where the sequences come in practically speaking, even modulo the aforementioned issues. How does the FVAE or its training scheme use this information? Does it use this information?

I think the positive experimental results in Figure 6c mean that there is something here. However, I cannot tell given section 4 what is actually being done.

Suggestions
====

I suggest a clear procedure section with numbered steps. It is of vital importance that the reader understand what has been done. If it is already there, it should be made much more obvious/clear.

I think the connection between sequences of images under actions and the proposed method needs to be made, or, if I missed this connection, should be made clear.

The initial portion of section 3 concerning dataset construction might also be moved to much later.

There are philosophically challenging sections which I did not comment on in the review portion. I think these are approximately orthogonal to the method due to the scope of the problem: rotation/translation may be disentangled. Can dog breeds *be* disentangled, even in theory (example from Section 3)? Disentanglement of simple mechanisms/"actions" are perfectly acceptable at least in my opinion for the state of the field at this moment. Using more complex examples may not be helpful. Similarly, the discussion in section 1 raises questions that are unrelated to the later method. Since a literature review is undertaken in Section 2, the paper could have started at "In this paper, we first demonstrate that instead of the ground-truth factors the disentangling approaches [should] learn [disentangled] actions.", with an update for phrasing.

I would also give the paper another read through for grammar.

---

### Author Response · Authors · 2020-11-24
**Response to All Reviewers**

I apologize for my poor writing skills and all the defects of our paper. It seems necessary to reclaim the motivation of this work. This paper's main purpose is to emphasize the importance of the data itself on disentanglement learning. We argue that a proper definition of disentanglement is disentangling explanatory action sequences because an isolating sample and its representation are insufficient for disentanglement or interpretability.  The internal relationships between the samples are the key to understanding and disentanglement.

The current works mainly try to interpret disentanglement from the model perspective.
1. AnnealedVAE claims that an information bottleneck enforces the model to find a local minimum for the objective, and "which are aligned with factors of variation." It suggests that increasing information on latent leads disentanglement.
2. Rolınek shows the similarity between PCA and VAE about "the local behavior of promoting both reconstruction and orthogonality."
However, we believe that the data plays a primary role, and the model is secondary. Therefore, we examine the effects of the data by four cases in Sec. 3. The special cases show little correlation between the learned representation and the ground-truth. Though PCA-like behaviors can interpret the A1-A3, it needs more explanations to interpret A4. The other difficulty is measuring the principal component quantitatively. Dividing the dataset into action sequences also helps us calculate the entropy of actions or the variation of components.

Our method is similar to AnnealedVAE both in results and the method; nevertheless, the interpretation is the main difference. We argue that the step values of beta instead of gradually modifying are vital to disentanglement. The other reason is the phenomenon of re-entanglement. The disentanglement metric reaches the highest in the middle phase, and it falls on the last phases. Here is the MIG score of one trail (beta-vae):

| Step | discrete_mig        |
|------|---------------------|
| 230  | 0.3756360504736129  |
| 461  | 0.37883395407382375 |
| 693  | 0.399240366607318*  |
| 924  | 0.3772557544010944  |
| 1156 | 0.3333029456260484  |
| 1387 | 0.3523813802097221  |
| 1618 | 0.3801707492631984  |
| 1850 | 0.349180837978194   |
| 2081 | 0.3663198905598549  |
| 2313 | 0.34369199263536326 |
| 2544 | 0.3663410570974127  |
| 2775 | 0.353156190268057   |

---

### Decision · Program_Chairs · 2021-01-07
**Final Decision**

**Decision:**

Reject

**Comment:**

The initial round of reviews showed a consensus among the reviewers that the presentation of the paper was poor, the novelty was unclear, claims were not properly justified, and the experimental evaluation and discussion were quite insufficient. The authors provided a rebuttal and an updated version of the paper. Although the updated paper demonstrated that the proposed approach indeed provides some benefits, it appears that the authors were not successful to address the numerous but constructive reviewers' comments.

The paper is not ready for publication in ICLR 2021 and can benefit from major revisions and careful proofreading.